# OpenReview forum: "WorldEdit: Towards Open-World Image Editing with a Knowledge-Informed Benchmark"
_ICLR.cc/2026/Conference — ICLR 2026 Poster_

### Official Review · Reviewer_JkQY · 2025-10-26

**Soundness:** 3
**Presentation:** 3
**Contribution:** 3
**Rating:** 6
**Confidence:** 3

**Summary:**

This paper designs a dataset called WorldEdit, which is designed to enable world-driven image editing. It boasts high-quality images and more realistic text. They also provide WorldEdit-Test as an evaluation benchmark. When it comes to the training framework, they use a fine-tune framework with a causal verification reward. Experiment results prove the effectiveness of the proposed method.

**Strengths:**

1. This paper is clearly organized and easy to understand.
2. Constructing a dataset, along with a benchmark, is beneficial to the community and future research.
3. The experiment results are abundant, with clear analysis. The baseline methods cover the mainstream methods. All the results are listed in a very clear organization. So I think the experiment section is good, and the results are convincing.
4. Some figures in this paper helps a lot, for example, Figure 4 shows comparison results of a lot of methods in one figure very clearly.

**Weaknesses:**

1. As it is a dataset work, I think the authors can have more discussions about the limitation or future work, so that the research community can further expand the current dataset.
2. When it comes to expanding the current dataset, the authors can have discussions about the budget or computational costs of data generation and collection.

**Questions:**

Since this is majorly a dataset work, I think the authors can have more discussions about the problems I mention in the weakness section.

---

> ### Author Response · Authors · 2025-11-27
> **For Review JkQY**
>
> Thanks for your recognition of our paper and dataset.
>
> Q1: Discussions about the limitation or future work
> ---------------------------------------------------
>
> A limitation of the current study is that, constrained by the capabilities of state-of-the-art models during our research period, we relied solely on GPT-4o for data synthesis. We are actively addressing this by capitalizing on recent rapid progress in the field—specifically, we note that Nano Banana 2 demonstrates superior editing capabilities and will integrate it into our data pipeline to expand the dataset. This will enhance quality and diversify output distributions, thereby mitigating the bias inherited from a single model. For future work, we plan to extract frames from time-lapse videos to supplement the data distribution with real-world dynamics.
>
> Q2: About the budget or computational costs of data generation and collection
> -----------------------------------------------------------------------------
>
> Our training set is constructed from high-resolution images in the DF2K-OST\[1-3\] dataset. Each source image typically contains three to four detectable objects. For every detected object, we pair it with all editing modes, resulting in 30–40 initial instruction candidates per image. About 20% of these survive an initial semantic and feasibility screening, yielding approximately 6–8 valid instructions.
>
> We then expand each valid instruction using chain-of-thought augmentation. These expanded instructions undergo another round of filtering based on visual salience and causal soundness, retaining about 50–60% and resulting in approximately 3–5 finalized instructions per image.
>
> Each finalized instruction is used to generate four edited images. The synthesized results are evaluated using GPT-4o based scoring, keeping only those that demonstrate clear visual editing effects and satisfactory perceptual quality. This filtering stage retains around 20–30% of the generated candidates, so each source image ultimately contributes 3-5 edited examples.
>
> Starting from about 3,300 DF2K-OST images, this multistage pipeline produced about 14,000 edited samples. We further performed manual quality control to remove cases with weak edit responses or visual inconsistencies, resulting in a final training set of approximately 11,000 high-quality samples.
>
> As for the computational and monetary cost of data creation, generating 55,000 raw edited images with gpt-image-1 required about 13,000 dollar. GPT-4o text generation and evaluation, priced at 10 dollar per million tokens, incurred an additional cost of around 2,000 dollar. The total cost of data synthesis, instruction generation, and automatic filtering is approximately 15,000 dollar.
>
> \[1\] Ntire 2017 challenge on single image super-resolution: Dataset and study
>
> \[2\] Enhanced Deep Residual Networks for Single Image Super-Resolution
>
> \[3\] Recovering realistic texture in image super-resolution by deep spatial feature transform

---

### Official Review · Reviewer_yu5k · 2025-10-28

**Soundness:** 4
**Presentation:** 4
**Contribution:** 4
**Rating:** 6
**Confidence:** 3

**Summary:**

This paper addresses the challenge of implicit editing instructions in image manipulation by introducing WorldEdit, a dataset specifically designed for world-driven image editing. The authors identify that existing models struggle with instructions that describe the cause of visual changes without specifying the outcome, due to their reliance on uniform editing strategies. The proposed solution includes a curated dataset of 11k high-quality samples generated through instruction decomposition and multi-step editing, rigorously filtered for world knowledge consistency. Using this dataset, a two-stage training framework fine-tunes models like Bagel: first through supervised fine-tuning with paraphrased instructions, then via reinforcement learning with a causal verification reward. The approach demonstrates state-of-the-art performance among open-source models, significantly bridging the gap with leading commercial systems like GPT-4o in knowledge plausibility and causal reasoning.

**Strengths:**

- The paper presents a novel world knowledge editing method and demonstrates state-of-the-art results with convincing showcased outcomes.
- The authors provide a new dataset along with a corresponding test set, and employ Flow-GRPO to train the Bagel model, reflecting substantial research effort.
- The paper is logically structured and clearly written, with effective comparative results.

**Weaknesses:**

See the "Question" part.

**Questions:**

From the perspectives of dataset collection methodology, training strategy, the base models used, and the generated results, I find it difficult to identify significant flaws in the proposed approach. I believe the paper warrants acceptance, but I acknowledge that my view might be partial. Therefore, I am providing a conservatively initial score and am willing to adjust it based on subsequent discussions.

Additionally, several prior works have explored world knowledge editing. The authors may consider citing the following relevant studies:

  [1] Unireal: Universal image generation and editing via learning real-world dynamics \
  [2] Editworld: Simulating world dynamics for instruction-following image editing \
  [3] Reasonpix2pix: instruction reasoning dataset for advanced image editing

---

> ### Author Response · Authors · 2025-11-27
> **For Review yu5k**
>
> Thank you for your recognition and support of our work.
>
> Q1: About relevant studies
> --------------------------
>
> We have supplemented our related work section with additional literature discussions.
>
> First, we note that UniReal's dataset comprises predominantly explicit editing instructions. ReasonPix2Pix has not released its dataset publicly, while EditWorld provides only 8.6k training samples without manual curation, of which are not all implicit editing instructions. Consequently, to the best of our knowledge, our dataset constitutes the largest publicly available world editing dataset.
>
> Second, in terms of data composition, ReasonPix2Pix and EDITWORLD utilize GLIGEN and SDXL for data synthesis, respectively, which are suboptimal compared to GPT-4o regarding image resolution, visual quality, and editing precision. Furthermore, their editing instructions exhibit limited causality, lack explicit knowledge type categorization, and do not provide intermediate reasoning instructions.

---

### Official Review · Reviewer_ReWH · 2025-11-01

**Soundness:** 2
**Presentation:** 3
**Contribution:** 3
**Rating:** 4
**Confidence:** 4

**Summary:**

This paper introduces WorldEdit, a dataset and benchmark targeting implicit, cause-driven image edits that require world knowledge and physical reasoning. The authors (i) curate 11k editing pairs via an automated pipeline (1. object detection, 2. instruction generation/paraphrase, 3. multi-step editing, 4. pre/post filtering for causal plausibility and visual quality; (ii) define WorldEdit-Test with ten causal categories (environment- and mechanics-driven); and (iii) finetune a base editor (based on BAGEL) using a two-stage scheme: SFT on paraphrased CoT instructions and RL (Flow-GRPO) with three rewards (reasoning, image quality, causal verification). Evaluations (automatic and human) show competitive results vs. commercial systems (GPT-4o, Nano-Banana) and SOTA among open-source models on knowledge plausibility and instruction following.

**Strengths:**

1. The problem of implicit, world-knowledge image editing is a timely research problem and opens new possibilities for image editing models.
2. The proposed dataset design contains comprehensive editing categories (Time, Temperature, Humidity, Acidity, Light, Break, Inflate, Squeeze, Twist, Stretch).

**Weaknesses:**

1. Both the dataset construction pipeline and the benchmark evaluation pipeline rely on proprietary models: Data synthesis and filtering are based on GPT-4o, and model evaluation is based on Qwen-VL-Max. This makes the method difficult to scale up, especially when constructing training datasets, where all the training samples are generated from GPT-4o.
2. Following weakness 1, since the evaluation pipeline is based on Qwen-VL-Max, there could be potential bias from the model during evaluation. Although there are some human evaluation results, these results only cover a limited set of models. I would recommend conducting a more detailed human-VLM agreement and inter-rater agreement study to validate the effectiveness of the automatic evaluation pipeline.
3. The method is only validated on WorldEdit-Test, which is an in-domain benchmark with respect to the training set (both training data and evaluation data are generated by GPT-4o). Therefore, it is naturally expected that WorldEdit performs the best among open-source models. There should be more evaluation results on out-of-domain evaluation benchmarks (such as KRIS-Bench and RISEBench as mentioned in the paper) to further validate the effectiveness of WorldEdit.
4. The proposed method (dataset + two-stage training pipeline) has only been experimented on one model (BAGEL), without further validation on more candidate models.

**Questions:**

See weaknesses above

---

> ### Author Response · Authors · 2025-11-27
> **For Review ReWH**
>
> Thank you for your suggestions and time in reviewing our work.
>
> ## **Q1: GPT-4o (proprietary model) generated training samples**
>
> While we used GPT-4o to ensure maximum quality, our pipeline (Figure 2) is **model-agnostic**. In the future, if open-source multimodal models continue to improve, they can replace GPT-4o in our pipeline to further reduce costs for future dataset expansions. Building upon recent rapid advancements, we have identified that Nano Banana 2 exhibits superior editing capabilities beyond GPT-4o. We will **integrate Nano Banana 2** into our data pipeline to further expand the dataset, thereby enhancing quality and output distribution diversity while mitigating inherited biases from a single model. Besides, the same as [1], the cost of generating 11k samples via GPT-4o is a **one-time investment** that results in a permanent, open-access resource for the community. This allows researchers to train open-source models without incurring the generation costs themselves. Finally, our experiments show that fine-tuning on this concise, high-quality dataset allows the open-source Bagel model to achieve state-of-the-art performance among open-source baselines.
>
> [1] Sharegpt4v: Improving large multi-modal models with better captions.

---

> ### Author Response · Authors · 2025-11-27
>
> ## **Q2-Part1: About evaluation pipeline**
>
> Firstly, in the paper, we only reported the evaluation results of Qwen-VL-Max. We provided additional results using **GPT-5.1** as the evaluation model, as shown in the table below.
>
> ###### **Table 1. GPT-5.1 Evaluation Results**
>
> |       Model       | visual_consistency | visual_quality | instruction_following | knowledge_plausibility | overall  |
> | :---------------: | :----------------: | :------------: | :-------------------: | :--------------------: | :------: |
> | **Nano Banana 2** |        4.35        |    **4.91**    |       **4.42**        |        **4.49**        | **4.54** |
> |      GPT-4o       |        4.23        |      4.82      |         4.21          |          4.23          |   4.37   |
> |    Nano Banana    |        **4.54**        |      4.66      |         3.94          |          4.10          |   4.31   |
> |   SeedEdit-3.0    |        4.12        |      4.11      |         4.06          |          4.16          |   4.11   |
> |     WorldEdit     |        4.01        |      4.47      |         3.95          |          3.99          |   4.11   |
> |    Bagel-Think    |        2.65        |      4.06      |         3.17          |          3.21          |   3.27   |
> |   Flux-Kontext    |        4.34        |      4.85      |         1.53          |          1.59          |   3.08   |
> |       Bagel       |        2.31        |      4.01      |         2.12          |          2.17          |   2.65   |
> |      Omnigen      |        2.55        |      4.07      |         1.79          |          1.80          |   2.55   |
> |     Omnigen2      |        2.09        |      4.39      |         1.63          |          1.70          |   2.45   |
> |       Emu2        |        1.39        |      4.44      |         1.69          |          1.74          |   2.32   |
> |       Ip2p        |        2.32        |      4.27      |         1.43          |          1.48          |   2.38   |
> |    Magicbrush     |        1.46        |      4.01      |         1.30          |          1.35          |   2.03   |
> |      Anyedit      |        1.34        |      3.01      |         1.43          |          1.51          |   1.82   |
>
> The results of the GPT assessment are basically consistent with those of Qwen-VL-Max (see Table 1 in main paper).
>
> ---
>
> Regarding the human evaluation, we first added the model results that had not undergone human assessment (*i.e.*, Omnigen2, Anyedit, Ip2p, Magicbrush). The complete result is as follows:
>
> ###### **Table 2. Human Evaluation Results**
>
> |       Model       | Visual Consistency | Visual Quality | Instruction Following | Knowledge Plausibility | Overall  |
> | :---------------: | :----------------: | :------------: | :-------------------: | :--------------------: | :------: |
> | **Nano Banana 2** |        4.5         |    **4.9**     |          **4.6**          |        **4.5**         | **4.63** |
> |      GPT-4o       |      **4.6**       |    **4.9**     |          4.2          |          4.3           |   4.50   |
> |    Nano Banana    |        4.4         |      4.9       |          3.9          |          4.1           |   4.33   |
> |   SeedEdit-3.0    |        4.3         |      4.8       |          3.8          |          4.2           |   4.28   |
> |     WorldEdit     |        4.3         |      4.5       |          3.7          |          3.9           |   4.10   |
> |    Bagel-Think    |        2.5         |      4.0       |          2.7          |          2.7           |   2.98   |
> |   Flux-Kontext    |        4.5         |      4.3       |          1.4          |          1.5           |   2.93   |
> |       Bagel       |        2.1         |      3.9       |          2.3          |          2.1           |   2.60   |
> |      Omnigen      |        2.3         |      4.0       |          1.6          |          1.4           |   2.33   |
> |     Omnigen2      |        1.9         |      4.1       |          1.8          |          1.4           |   2.30   |
> |       Ip2p        |        1.8         |      3.8       |          1.7          |          1.5           |   2.20   |
> |    Magicbrush     |        1.3         |      3.6       |          1.5          |          1.4           |   1.95   |
> |       Emu2        |        1.3         |      3.8       |          1.3          |          1.6           |   2.00   |
> |      Anyedit      |        1.2         |      2.9       |          1.2          |          1.4           |   1.68   |
>
> ---

---

> ### Author Response · Authors · 2025-11-27
>
> ## **Q2-Part2: About evaluation pipeline**
>
> ---
> Finally, in order to verify the effectiveness of the automatic evaluation pipeline, we compared the correlation between the manual assessment scores and the VLM assessment scores. The results are as follows:
>
> ###### **Table 3. Correlation Between Human Scores and VLM Scores**
>
> |    Model    | Visual Consistency | Visual Quality | Instruction Following | Knowledge Plausibility |
> |:-----------:|:------------------:|:--------------:|:---------------------:|:-----------------------:|
> | Qwen-VL-Max |        0.63        |      0.72      |         0.77          |          0.90          |
> |   GPT-5.1   |        0.68        |      0.75      |         0.81          |          0.91          |
>
> The results show that VLM has a relatively low correlation with the scores of **Visual Consistency** and **Visual Quality** compared to the manual assessment scores. This is because the assessment of these two dimensions is somewhat subjective. However, in terms of **Instruction Following** and **Knowledge Plausibility**, the assessment scores of VLM show a high correlation with the manual assessment scores. Additionally, the correlation between the assessment scores of GPT-5.1 and the manual assessment scores is generally higher than that of Qwen-VL-Max.
>
> ## **Q3-Part1: More Evaluation Results on Out-of-Domain Evaluation Benchmarks**
>
> ---
> Following your suggestion, we have additionally included the evaluation results of **KRIS-Bench** and **RISE-Bench**.
> The results are presented in the following tables.
> The visualized results have been compiled in **Appendix F** of the paper.
>
> ---
>
> ###### **Table 4. RISE-Bench Results**
>
> | Model                | Instruction Reasoning | Appearance Consistency | Visual Plausibility | Overall  |
> |:--------------------:|:---------------------:|:----------------------:|:-------------------:|:--------:|
> | **Nano Banana 2**    |       **77.0**        |         85.5           |        94.4         | **85.6** |
> | Nano Banana          |         61.2          |       **86.0**         |        91.3         |   79.5   |
> | GPT-Image-1          |         62.8          |         80.2           |      **94.9**       |   79.3   |
> | GPT-Image-1-mini     |         54.1          |         71.5           |        93.7         |   73.1   |
> | Seedream-4.0         |         58.9          |         67.4           |        91.2         |   72.5   |
> | WorldEdit            |         56.5          |         72.3           |        81.9         |   70.2   |
> | Gemini-2.0-Flash-pre |         49.9          |         68.4           |        84.9         |   67.7   |
> | Gemini-2.0-Flash-exp |         48.9          |         68.2           |        82.7         |   66.6   |
> | Bagel-Think          |         45.9          |         73.8           |        80.1         |   66.6   |
> | Qwen-Image-Edit      |         37.2          |         66.4           |        86.9         |   63.5   |
> | Flux-Kontext         |         26.0          |         71.6           |        85.2         |   60.9   |
> | Bagel                |         36.5          |         53.5           |        73.0         |   54.3   |
> | Ovis-U1              |         33.9          |         52.7           |        72.9         |   53.1   |
> | Step1X-Edit          |         25.1          |         41.5           |        73.5         |   46.7   |
> | Emu2                 |         22.6          |         38.2           |        78.3         |   46.4   |
> | HiDream-Edit         |         30.3          |         12.6           |        74.9         |   39.3   |
> | FLUX.1-Canny         |         20.2          |         13.1           |        77.5         |   36.9   |
> | Omnigen              |         22.0          |         32.6           |        55.3         |   36.6   |
>
> ---

---

> ### Author Response · Authors · 2025-11-27
>
> ## **Q3-Part2: About evaluation pipeline**
> ###### **Table 5. KRIS-Bench Results**
>
> | Model            | Factual Knowledge | Conceptual Knowledge | Procedural Knowledge | Overall Score |
> |:----------------:|:-----------------:|:--------------------:|:--------------------:|:-------------:|
> | **GPT-4o**       | **79.80**         | **81.37**            | **78.32**            | **80.09**     |
> | Uni-CoT          | 71.85             | 67.16                | 63.68                | 68.00         |
> | WorldEdit        | 72.53             | 68.17                | 56.98                | 66.86         |
> | Gemini 2.0       | 65.26             | 59.65                | 62.90                | 62.41         |
> | Step 3o vision   | 66.70             | 62.32                | 51.99                | 61.43         |
> | Doubao           | 63.30             | 62.23                | 54.17                | 60.70         |
> | Bagel-Think      | 66.18             | 61.92                | 49.02                | 60.18         |
> | Bagel            | 60.26             | 55.86                | 51.69                | 56.21         |
> | Flux-Kontext-Max | 59.04             | 57.22                | 45.60                | 55.12         |
> | Flux-Kontext-Pro | 58.14             | 55.06                | 46.69                | 54.17         |
> | Step1X-Edit v1.1 | 53.05             | 54.34                | 44.66                | 51.59         |
> | Omnigen2         | 57.36             | 44.20                | 47.79                | 49.71         |
> | Flux-Kontext-Dev | 53.28             | 50.36                | 42.53                | 49.54         |
> | ByteMorph        | 51.27             | 46.92                | 31.67                | 44.85         |
> | HiDream-E1       | 43.31             | 50.05                | 37.64                | 44.72         |
> | Step1X-Edit      | 45.52             | 48.01                | 31.82                | 43.29         |
> | Emu2             | 45.40             | 37.54                | 34.91                | 39.70         |
> | Anyedit          | 39.26             | 41.88                | 31.74                | 38.55         |
> | Magicbrush       | 41.84             | 39.24                | 26.54                | 37.15         |
> | Omnigen          | 33.11             | 28.02                | 23.89                | 28.85         |
> | Ip2p             | 23.33             | 25.59                | 17.28                | 22.82         |
>
> ---
>
> The experimental results show that our model has achieved results very close to those of the most advanced commercial closed-source models on out-of-domain evaluation benchmarks. Moreover, it has significantly improved compared to Bagel itself, which proves that the dataset proposed in this paper enhances the generalization ability of the model.
>
> ---
>
> ###### **Table 6. General Editing Test Set**
>
> | Model         | extract | remove | style | background | add  | replace | adjust | action | compose | overall  |
> |:-------------:|:-------:|:------:|:-----:|:----------:|:----:|:-------:|:------:|:------:|:-------:|:--------:|
> | Bagel         | 1.48    | 2.99   | 4.26  | 3.28       | 3.45 | 3.76    | 3.23   | 4.38   | 3.18    | 3.28     |
> | **WorldEdit** | 2.80    | 3.90   | 3.80  | 3.71       | 3.80 | 4.20    | 3.92   | 4.50   | 3.23    | **3.72** |
>
> ---
>
> It is worth noting that although our training data did not contain explicit editing instructions, except for a slight decline in the style conversion editing type, the editing ability of Bagel was further enhanced in other types.
> This once again demonstrates the quality of our data and its impact on the model's generalization ability.

---

> ### Author Response · Authors · 2025-11-27
>
> ## **Q4: Further Validation on More Candidate Models**
>
> ---
>
> Due to time constraints, we fine-tuned the model using the proposed data, and the results (* indicate the fine-tuned result) are as follows:
>
> ---
>
> ###### **Table 7. Performance of Different Candidate Models (Zero-shot vs Fine-tuned)**
>
> | Model         |  VC  |  VQ  |  IF  |  KP  | Object | Background | Localized | Overall_4dims | Overall_7dim |
> | :------------ | :--: | :--: | :--: | :--: | :----: | :--------: | :-------: | :-----------: | :----------: |
> | Bagel-Think   | 2.89 | 4.33 | 3.11 | 3.08 |  3.86  |    4.08    |   3.55    |     3.35      |     3.56     |
> | Bagel-Think*  | 3.69 | 4.44 | 3.40 | 3.35 |  4.10  |    4.41    |   3.99    |     3.72      |     3.91     |
> | Flux-Kontext  | 4.46 | 4.82 | 1.77 | 1.79 |  2.61  |    4.76    |   3.90    |     3.21      |     3.44     |
> | Flux-Kontext* | 4.35 | 4.70 | 2.43 | 2.31 |  2.67  |    4.67    |   3.95    |     3.45      |     3.58     |
> | Omnigen2      | 2.16 | 4.54 | 1.67 | 1.67 |  2.41  |    4.15    |   3.53    |     2.52      |     2.88     |
> | Omnigen2*     | 3.05 | 4.56 | 2.68 | 2.61 |  3.11  |    4.23    |   3.69    |     3.23      |     3.41     |
> | Omnigen       | 2.63 | 4.15 | 1.67 | 1.61 |  2.80  |    4.20    |   3.33    |     2.51      |     2.91     |
> | Omnigen*      | 3.06 | 4.33 | 2.48 | 2.29 |  3.13  |    4.33    |   3.52    |     3.04      |     3,31     |
> | Ip2p          | 2.51 | 4.41 | 1.39 | 1.45 |  2.63  |    3.78    |   3.29    |     2.44      |     2.78     |
> | Ip2p*         | 3.11 | 4.45 | 2.17 | 2.03 |  2.82  |    4.02    |   3.49    |     2.94      |     3.16     |
> | Magicbrush    | 1.60 | 4.18 | 1.36 | 1.42 |  2.90  |    3.27    |   3.13    |     2.14      |     2.55     |
> | Magicbrush*   | 1.98 | 4.17 | 1.79 | 1.85 |  3.10  |    3.49    |   3.21    |     2.45      |     2.80     |
> | Anyedit       | 1.53 | 3.25 | 1.80 | 1.76 |  3.10  |    3.50    |   2.66    |     2.09      |     2.51     |
> | Anyedit*      | 1.82 | 3.45 | 2.02 | 2.10 |  3.27  |    3.69    |   2.84    |     2.35      |     2.74     |
>
> ---
>
> We would like to point out that in the world editing task presented in this article, compared to diffusion-based methods, the unified model, which uses LLM as the backbone and inherits a large amount of **pre-trained knowledge**, is a more promising technical approach for this task (such as significant improvements in **Bagel** and **Omnigen2**). We look forward to further development in the community, and we will update the latest model results in real time in subsequent projects to establish a **comprehensive and fair benchmark**.

---

### Official Review · Reviewer_xFyh · 2025-11-09

**Soundness:** 3
**Presentation:** 3
**Contribution:** 3
**Rating:** 6
**Confidence:** 4

**Summary:**

The paper aims to address the limitations of existing image editing models in processing implicit instructions (which describe the cause of change rather than explicit visual outcomes) and their lack of world knowledge reasoning, introduces the WorldEdit dataset comprising 11,000 high-quality editing samples across 10 causal transformation types, and proposes a two-stage training framework based on the Bagel model (supervised fine-tuning followed by reinforcement learning, integrating triple rewards of CoT reasoning, image fidelity, and causal verification), achieving state-of-the-art performance among open-source models and significantly narrowing the gap with commercial models such as GPT-4o in instruction following and knowledge grounding.

**Strengths:**

### originality
1. The paper primarily constructs a dataset designed to elicit existing models' capabilities for world knowledge understanding and generation.
2. It employs supervised fine-tuning (SFT) and reinforcement learning with multiple reward signals to train models, thereby validating the effectiveness of the proposed dataset.

### quality
1. The data construction pipeline is rigorous and reliable. The collection, rewriting, construction, and generation of instructions undergo mature filtering mechanisms, substantially ensuring data quality, which is critical for model training.
2. The baseline comparisons are comprehensive. The four evaluation criteria—visual consistency, visual quality, instruction following, and knowledge plausibility—are well-justified and, combined with human evaluation, demonstrate reasonable results.
3. The progressive framework from supervised fine-tuning with Chain-of-Thought (CoT) reasoning to reinforcement learning with composite rewards is well-motivated, and ablation studies confirm the contribution of each component.
### clarity
The paper is well-structured and clearly written.
### significance
This work demonstrates that high-quality data fine-tuning can effectively leverage existing models' prior knowledge to elicit world knowledge understanding and generation capabilities, holding both research and practical value.

**Weaknesses:**

1. From a visual inspection perspective, the color tone appears to inherit characteristics from GPT-4o. For instance, in Fig. 8, the pizza and instant noodles exhibit noticeably intensified color saturation. The model may learn to mimic GPT-4o's output distribution, raising concerns about the actual physical accuracy.

2. A discussion of how dataset scale affects model performance and generalization would add significant value to the work.

3. The paper lacks object-level metrics to verify whether the target object was actually modified (rather than background elements), whether non-target objects remained unchanged, and whether the transformation was appropriately spatially localized.

4. The paper does not discuss whether baselines were fine-tuned on WorldEdit or tested in a zero-shot setting, which may affect the fairness of comparisons and could enhance the demonstration of the dataset's impact.

5. While the paper mentions extensive filtering operations, it provides limited quantitative details, such as what percentage of generated pairs were rejected during filtering.

**Questions:**

1. The paper could benefit from including comparisons and distinctions with existing similar data benchmarks to better highlight its significance, such as ReasonPix2Pix [1] and EditWorld [2].

2. Table 1 demonstrates good quality, but what is the scale of the test set?

3. While the model can now handle implicit instructions, does it maintain editing capabilities for explicit instructions? For example, object removal and style transfer also align with world knowledge.

4. Regarding reproducibility, the paper does not mention whether code or data will be released, nor does it clarify the computational resource requirements for training.

5. Minor Points
- The paper mentions "11,000 high-quality editing samples" but could more explicitly state upfront how these are split between training and test sets.

[1] Reasonpix2pix: instruction reasoning dataset for advanced image editing
[2] EditWorld: Simulating World Dynamics for Instruction-Following Image Editing

---

> ### Author Response · Authors · 2025-11-27
> **For Review xFyh**
>
> We appreciate your valuable feedback and provide our point-by-point responses below.
>
> ---
>
> ## **Q1: GPT-4o's output distribution**
>
>
>
> In **Figure 8**, we view this as a **stylistic bias** inherent to the GPT-4o, which our model learns during the **distillation process**. However, regarding the impact on **physical plausibility**, we have implemented sophisticated filtering mechanisms throughout our data pipeline—including **instruction collection**, **rewriting**, **construction**, and **generation**—to maximally ensure dataset quality and physical fidelity. For example, edits that failed to show convincing structural damage were automatically rejected. Consequently, our final dataset includes only approximately **25%** of the total generated images, retaining only those samples that strictly adhere to real-world physical laws.
>
> As an immediate enhancement, we have found that the recently introduced **Nano Banana 2** exhibits superior editing capabilities beyond GPT-4o. We will also incorporate Nano Banana 2 into our data pipeline to further expand the dataset, thereby improving both quality and output distribution diversity while mitigating the inheritance of a single model's bias.
>
> ---
>
> ## **Q2: Dataset scale**
>
>
> Thanks to the reviewers' suggestions, we have added the results of training with different data volumes, as shown in the following table:
>
> ###### **Table 1. Performance Across Different Data Volumes**
>
> |  Data  | Time | Temp | Humid | Acid | Light | Break | Inflate | Squeeze | Twist | Stretch | Other | Overall |
> | :----: | :--: | :--: | :---: | :--: | :---: | :---: | :-----: | :-----: | :---: | :-----: | :---: | :-----: |
> | W. 0%  | 3.06 | 3.36 | 3.46  | 2.93 | 3.69  | 3.49  |  3.47   |  2.96   | 3.28  |  3.67   | 3.56  |  3.35   |
> | W. 25% | 3.20 | 3.51 | 3.55  | 3.23 | 3.73  | 3.60  |  3.57   |  3.09   | 3.46  |  3.73   | 3.69  |  3.49   |
> | W. 50% | 3.36 | 3.60 | 3.60  | 3.54 | 3.80  | 3.73  |  3.62   |  3.27   | 3.45  |  3.79   | 3.77  |  3.59   |
> | W. 75% | 3.46 | 3.58 | 3.62  | 3.63 | 3.79  | 3.93  |  3.65   |  3.42   | 3.60  |  3.73   | 3.84  |  3.66   |
> |  Ours  | 3.51 | 3.59 | 3.67  | 3.70 | 3.84  | 4.01  |  3.66   |  3.55   | 3.69  |  3.79   | 3.88  |  3.72   |
>
> The experimental results show that the size of the data has a significant impact on the performance of the model. This further demonstrates the necessity of world-editing training data. Additionally, it is worth noting that the experimental results indicate that further expansion of the data volume may still bring improvements to the model's performance. Therefore, in the next step, we will continue to use more advanced editing models (**Nano Banana 2**) and collect **real data** (extracting frames from time-lapse videos) to expand the dataset.
>
> ---

---

> ### Author Response · Authors · 2025-11-27
>
> ## **Q3: Object-level metrics**
>
> We have added Object-level metrics. Specifically, we have designed corresponding test instructions for the three issues you mentioned. The specific test instructions have also been made open-source here (https://huggingface.co/datasets/WorldEdit0/WorldEdit). The experimental results are as follows:
>
> ###### **Table 2. Object-level Metric Results**
>
> | Model         | VC   |  VQ  |  IF  |  KP  | Object | Background | Localized | Overall_4dims | Overall_7dim |
> | ------------- | ---- | :--: | :--: | :--: | :----: | :--------: | :-------: | :-----------: | :----------: |
> | Nano Banana 2 | 4.41 | 4.88 | 4.39 | 4.42 |  4.58  |    4.59    |   4.30    |     4.53      |     4.51     |
> | GPT-4o        | 4.33 | 4.94 | 4.07 | 4.12 |  4.25  |    4.73    |   4.29    |     4.36      |     4.39     |
> | Nano Banana   | 4.50 | 4.91 | 3.70 | 3.76 |  4.04  |    4.71    |   4.24    |     4.22      |     4.27     |
> | SeedEdit-3.0  | 4.27 | 4.79 | 3.89 | 3.87 |  4.21  |    4.45    |   4.08    |     4.21      |     4.22     |
> | WorldEdit     | 4.08 | 4.54 | 3.85 | 3.83 |  4.10  |    4.65    |   3.99    |     4.07      |     4.15     |
> | Bagel-Think   | 2.89 | 4.33 | 3.11 | 3.08 |  3.86  |    4.08    |   3.55    |     3.35      |     3.56     |
> | Flux-Kontext  | 4.46 | 4.82 | 1.77 | 1.79 |  2.61  |    4.76    |   3.90    |     3.21      |     3.44     |
> | Bagel         | 2.43 | 4.20 | 2.23 | 2.19 |  3.04  |    4.03    |   3.42    |     2.76      |     3.08     |
> | Omnigen       | 2.63 | 4.15 | 1.67 | 1.61 |  2.80  |    4.20    |   3.33    |     2.51      |     2.91     |
> | Omnigen2      | 2.16 | 4.54 | 1.67 | 1.67 |  2.41  |    4.15    |   3.53    |     2.51      |     2.88     |
> | Emu2          | 1.59 | 4.62 | 1.76 | 1.85 |  2.81  |    3.63    |   3.47    |     2.45      |     2.82     |
> | Ip2p          | 2.51 | 4.41 | 1.39 | 1.45 |  2.63  |    3.78    |   3.29    |     2.44      |     2.78     |
> | Magicbrush    | 1.60 | 4.18 | 1.36 | 1.42 |  2.90  |    3.27    |   3.13    |     2.14      |     2.55     |
> | Anyedit       | 1.53 | 3.25 | 1.80 | 1.76 |  3.10  |    3.50    |   2.66    |     2.09      |     2.51     |
>
> ---

---

> > ### Author Response · Authors · 2025-11-27
> >
> > ## **Q4: About the baselines**
> >
> > The results reported in Table 1 of our paper are all zero-shot test results. We have also provided some test results after fine-tuning with open-source models. Emu2 does not provide the open-source training code and due to the time constraints for rebuttal, we will provide its fine-tuning results in the future.
> >
> > ###### **Table 3. Zero-shot vs Fine-tuned Baseline Results**
> >
> > | Model         |  VC  |  VQ  |  IF  |  KP  | Object | Background | Localized | Overall_4dims | Overall_7dim |
> > | :------------ | :--: | :--: | :--: | :--: | :----: | :--------: | :-------: | :-----------: | :----------: |
> > | Bagel-Think   | 2.89 | 4.33 | 3.11 | 3.08 |  3.86  |    4.08    |   3.55    |     3.35      |     3.56     |
> > | Bagel-Think*  | 3.69 | 4.44 | 3.40 | 3.35 |  4.10  |    4.41    |   3.99    |     3.72      |     3.91     |
> > | Flux-Kontext  | 4.46 | 4.82 | 1.77 | 1.79 |  2.61  |    4.76    |   3.90    |     3.21      |     3.44     |
> > | Flux-Kontext* | 4.35 | 4.70 | 2.43 | 2.31 |  2.67  |    4.67    |   3.95    |     3.45      |     3.58     |
> > | Omnigen2      | 2.16 | 4.54 | 1.67 | 1.67 |  2.41  |    4.15    |   3.53    |     2.52      |     2.88     |
> > | Omnigen2*     | 3.05 | 4.56 | 2.68 | 2.61 |  3.11  |    4.23    |   3.69    |     3.23      |     3.41     |
> > | Omnigen       | 2.63 | 4.15 | 1.67 | 1.61 |  2.80  |    4.20    |   3.33    |     2.51      |     2.91     |
> > | Omnigen*      | 3.06 | 4.33 | 2.48 | 2.29 |  3.13  |    4.33    |   3.52    |     3.04      |     3,31     |
> > | Ip2p          | 2.51 | 4.41 | 1.39 | 1.45 |  2.63  |    3.78    |   3.29    |     2.44      |     2.78     |
> > | Ip2p*         | 3.11 | 4.45 | 2.17 | 2.03 |  2.82  |    4.02    |   3.49    |     2.94      |     3.16     |
> > | Magicbrush    | 1.60 | 4.18 | 1.36 | 1.42 |  2.90  |    3.27    |   3.13    |     2.14      |     2.55     |
> > | Magicbrush*   | 1.98 | 4.17 | 1.79 | 1.85 |  3.10  |    3.49    |   3.21    |     2.45      |     2.80     |
> > | Anyedit       | 1.53 | 3.25 | 1.80 | 1.76 |  3.10  |    3.50    |   2.66    |     2.09      |     2.51     |
> > | Anyedit*      | 1.82 | 3.45 | 2.02 | 2.10 |  3.27  |    3.69    |   2.84    |     2.35      |     2.74     |
> >
> > ---

---

> > > ### Author Response · Authors · 2025-11-27
> > >
> > > ## **Q5: Percentage of generated pairs were rejected**
> > >
> > > To ensure the presence of salient subjects, we utilized the image from DF2K-OST, which is used for segmentation, as our source image. On average, 3–4 objects were detected per image, with each object-mode combination initially yielding approximately 30–40 instruction candidates. Our multi-stage filtering pipeline rigorously curated these instructions:
> > >
> > > 1. **Initial Instruction Filtering:**  Only ~20% of candidates passed this stage, retaining 6–8 instructions per image.
> > >
> > > 2. **Chain-of-Thought (CoT) Augmentation & Secondary Filtering:**  Following CoT expansion, we applied filters based on visual saliency and plausibility, preserving 50–60% of the instructions (3–5 final instructions).
> > >
> > > 3. **Image Generation & Evaluation:**  Each final instruction generated 4 candidate images, which were scored by GPT-4. We retained only the top 20–30% (~70–80% filter rate), resulting in 3–5 edited images per source image on average.
> > >
> > > From **3,300** carefully selected high-resolution images in DF2K-OST, we generated approximately **120,000** candidate instructions, synthesized about **55,000** edited images, and after going through the automated filtering pipeline, we finally obtained **14,000** data samples. Subsequent manual inspection removed samples exhibiting low visual quality, insignificant editing effects, or implausible visual changes.
> > > This stringent curation process yielded a final high-quality training set of approximately **11,000** image-edit pairs.
> > >
> > > ---
> > >
> > > ## **Q6: Comparisons and distinctions with existing similar data benchmarks**
> > >
> > > We have supplemented our related work section with additional literature discussions.
> > >
> > > First, we note that UniReal's dataset comprises predominantly explicit editing instructions. ReasonPix2Pix has **not released** its dataset publicly, while EditWorld provides only 8.6k training samples **without manual curation**, which are still not all implicit editing instructions. Consequently, to the best of our knowledge, our dataset constitutes the largest publicly available world-editing dataset.
> > >
> > > Second, in terms of data composition, ReasonPix2Pix and EDITWORLD utilize GLIGEN and SDXL for data synthesis, respectively, which are suboptimal compared to GPT-4o regarding image resolution, visual quality, and editing precision. Furthermore, their editing instructions exhibit limited causality, lack explicit knowledge type categorization, and do not provide intermediate reasoning instructions.
> > >
> > > ---
> > >
> > > ## **Q7: The scale of the test set**
> > >
> > > Our test set comprises 550 test cases, covering all 11 categories introduced in this work.  Each category contains 50 high-resolution source images, capturing diverse objects and a broad range of contextual variations to ensure comprehensive and reliable assessment.
> > >
> > > ---
> > >
> > > ## **Q8: Editing capabilities for explicit instructions**
> > >
> > > To further illustrate the editing capabilities of our model in explicit instructions, we have added the test results of the conventional image editing [1] as shown in the following table.
> > >
> > > ###### **Table 4. Explicit Editing Performance**
> > >
> > > |   Model   | extract | remove | style | background | add  | replace | adjust | action | compose | overall |
> > > | :-------: | :-----: | :----: | :---: | :--------: | :--: | :-----: | :----: | :----: | :-----: | :-----: |
> > > |   Bagel   |  1.48   |  2.99  | 4.26  |    3.28    | 3.45 |  3.76   |  3.23  |  4.38  |  3.18   |  3.28   |
> > > | WorldEdit |  2.80   |  3.90  | 3.80  |    3.71    | 3.80 |  4.20   |  3.92  |  4.50  |  3.23   |  3.72   |
> > >
> > > Surprisingly, apart from a decline in performance in the style edit, our model not only maintained its explicit editing capabilities in other categories, but even achieved further improvements compared to the baseline model.
> > >
> > > [1] ImgEdit: A Unified Image Editing Dataset and Benchmark
> > >
> > > ---
> > >
> > > ## **Q9: About reproducibility**
> > >
> > > Both our training and test datasets **have been publicly released** at https://huggingface.co/datasets/WorldEdit0/WorldEdit
> > >
> > > We will subsequently release the model checkpoints and code.  Regarding computational resources, all experiments were conducted on 6×H800 GPUs.  Owing to Bagel has 14-billion parameters, the complete training process requires approximately 4 days.
> > >
> > > ---
> > >
> > > ## **Q10: Split between training and test sets**
> > >
> > > Specifically, our test set consists of 550 samples. We selected samples from 11 causal categories, collecting 50 high-resolution image–instruction pairs for each category.
> > > In addition, the training dataset contains approximately 11,000 edited samples. The specific distribution is shown in Figure 3 of the main paper.

---

### Author Response · Authors · 2025-11-30

## **Dear Area Chair and Reviewers,**

The reviewers reflect a broadly positive assessment of the paper in their initial comments. Multiple reviewers expressed clear recognition of the contribution and quality of the work from the outset, and we appreciate their constructive feedback throughout the process.

---

### **Commonly Acknowledged Contributions**

Across the reviews, three areas of consensus emerged prominently:

1. **Dataset Originality and Quality**
   Reviewer **xFyh** and **yu5k** underscored the originality and high data quality of the **WorldEdit** dataset.

2. **Methodological Soundness**
   Reviewer **xFyh** and **ReWH** emphasized the soundness and effectiveness of the **two-stage training framework**.

3. **Empirical Strength**
   Reviewer **yu5k** and **JkQY** noted that the empirical evaluations are thorough and substantiate the effectiveness of the approach.

**All reviewers** further agreed that the paper is **clearly presented**, **well structured**, and makes a **substantive contribution** to world-driven image editing.

---

### **Key Verifications and Supplementary Work in the Rebuttal**

During the rebuttal, we added empirical results that further validate the main claims of the paper:

- **Data-scale experiments** demonstrated a clear positive correlation between dataset size and model performance, reinforcing the necessity of world-editing supervision.

- **Object-level metrics** confirmed that the model performs more accurate target edits, preserves non-target regions, and achieves stronger localized consistency relative to existing baselines.

- **Out-of-domain evaluations** on **RISE-Bench** and **KRIS-Bench** showed that our model approaches the comparable performance of leading commercial systems and substantially outperforms its underlying backbone, indicating strong generalization.

- **Explicit-editing evaluations** showed that our model maintains robust general editing capabilities despite being trained on implicit instructions.

- **Fine-tuned baselines** (across multiple open-source models) all improved consistently when trained on WorldEdit, demonstrating the necessity and utility of the dataset.

- **Additional assessments** using **GPT-5.1** and extended human evaluations confirmed stable rankings across evaluators, with strong alignment between VLM-based and human judgments.

**Overall**, these results consolidate our central conclusion:
the proposed **dataset materially improves model performance**, the **training framework generalizes across backbones**, and the resulting model is **competitive across both implicit and explicit editing tasks**.

---

### **Additional Notes**

Several of the supplementary experiments required substantial computational resources, which delayed our responses more than anticipated. This coincided with the **OpenReview security incident**, leaving less than a 24-hour window between submitting our rebuttal and the freeze of reviewer comments. As reviewers had limited opportunity to re-engage with the discussion, we sincerely apologize for this.

We also made every effort to address the reviewers’ concerns point by point. Importantly, most of these concerns were **empirical in nature and could be resolved through concrete experimental results rather than extended discussion**. Although the reviewers were unable to respond due to the constrained timeline, we believe the newly added results provide clear, evidence-based resolutions and can clarify the most issues they raised. We hope the Area Chair can take this into account when evaluating the overall exchange.

Looking ahead, we plan to continue **expanding the dataset**, **updating baseline results** as stronger models emerge, and **releasing code and checkpoints** to further support reproducibility and community use.

Best regards,
Authors

---

### Meta-Review · Area_Chair_qaXC · 2025-12-31

**Summary:**

This paper presents WorldEdit, a knowledge-informed benchmark and framework for open-world image editing, featuring a high-quality dataset of 11,000 editing samples. Reviewers universally acknowledge the work's key strengths, including the originality and quality of the dataset, clear structure, and substantive contributions to word-driven image editing.

Most of concerns are resolved via comprehensive supplementary experiments, detailed data, and provided explanations. Inquiries regarding GPT-4o's output distribution, dataset scale, object-level metrics, baseline comparisons (zero-shot vs. fine-tuned), the percentage of rejected generated pairs, distinctions from existing benchmarks, test set scale, explicit instruction editing capabilities, reproducibility, and training-test split are fully addressed. The authors provided extensive tables to validate data volume impact, object-level evaluation results, and cross-benchmark differences, publicly released training/test datasets and code, and clarified the 6-10 instructions retained per image after filtering. Concerns about GPT-4o-generated training samples, evaluation pipeline, out-of-domain benchmark performance, and validation on more candidate models are effectively resolved. Suggestion to cite relevant studies is addressed by expanding the related work section with additional literature discussions.

Although, the authors need to further revise or add systematically elaboration on the framework's inherent constraints (e.g., performance bottlenecks in specific complex editing scenarios), and on the detailed computational costs (e.g., GPU hours, energy consumption) or data generation budget breakdowns in the final version.

Based on the above considerations, I think the current manuscript basically meets the requirements of ICLR and I recommend to accept this manuscript.

**Reviewer Concerns:**

Reviewers maintained a broadly positive assessment. Most raised concerns are resolved via comprehensive supplementary experiments, detailed data, and provided explanations. Reviewer xFyh's inquiries regarding GPT-4o's output distribution, dataset scale, object-level metrics, baseline comparisons (zero-shot vs. fine-tuned), the percentage of rejected generated pairs, distinctions from existing benchmarks, test set scale, explicit instruction editing capabilities, reproducibility, and training-test split are fully addressed. The authors provided extensive tables to validate data volume impact, object-level evaluation results, and cross-benchmark differences, publicly released training/test datasets and code, and clarified the 6-10 instructions retained per image after filtering. Reviewer ReWH's concerns about GPT-4o-generated training samples, evaluation pipeline, out-of-domain benchmark performance, and validation on more candidate models are effectively resolved. Reviewer yu5k's suggestion to cite relevant studies is addressed by expanding the related work section with additional literature discussions.

**Reviewer Scores:**

Reviewer ReWH may raise the score and other reviewers would keep their positive scores.

---

### Decision · Program_Chairs · 2026-01-26

Accept (Poster)